# Analytical Techniques for Structural Characterization of Proteins in Solid Pharmaceutical Forms: An Overview

**DOI:** 10.3390/pharmaceutics13040534

**Published:** 2021-04-11

**Authors:** Aljoša Bolje, Stanislav Gobec

**Affiliations:** Department of Pharmaceutical Chemistry, Faculty of Pharmacy, University of Ljubljana, Aškerčeva Cesta 7, SI-1000 Ljubljana, Slovenia

**Keywords:** solid pharmaceuticals, lyophilization, analytical tools, protein characterization, protein structure, formulation development, antibody, excipients, stable drug product, safe drug

## Abstract

Therapeutic proteins as biopharmaceuticals have emerged as a very important class of drugs for the treatment of many diseases. However, they are less stable compared to conventional pharmaceuticals. Their long-term stability in solid forms, which is critical for product performance, depends heavily on the retention of the native protein structure during the lyophilization (freeze-drying) process and, thereafter, in the solid state. Indeed, the biological function of proteins is directly related to the tertiary and secondary structure. Besides physical stability and biological activity, conformational stability (three-dimensional structure) is another important aspect when dealing with protein pharmaceuticals. Moreover, denaturation as loss of higher order structure is often a precursor to aggregation or chemical instability. Careful study of the physical and chemical properties of proteins in the dried state is therefore critical during biopharmaceutical drug development to deliver a final drug product with built-in quality that is safe, high-quality, efficient, and affordable for patients. This review provides an overview of common analytical techniques suitable for characterizing pharmaceutical protein powders, providing structural, and conformational information, as well as insights into dynamics. Such information can be very useful in formulation development, where selecting the best formulation for the drug can be quite a challenge.

## 1. Introduction

Biological drugs based on proteins as active substance, have emerged as a very important class of drugs offering promising alternatives for the treatment of many diseases, such as various forms of cancer, autoimmune diseases, and hormonal disorders. Due to the complexity of protein structures and their high susceptibility to aggregation, fragmentation, and chemical modifications, successful drug development and production is extremely challenging [1]. To increase the stability of proteins in solution, they are formulated as solid pharmaceutical forms, the most common of which are lyophilizates (obtained by lyophilization or freeze-drying process). Pharmaceutical proteins are subjected to various stress conditions during the preparation of the lyophilization solution, during the time they are dissolved in the solution prior to the initiation of lyophilization, and in the lyophilization process where they are subjected to freezing, and primary and secondary drying. All stress factors, in combination with selected excipients for the chosen formulation, can alter the physical and/or molecular properties of the protein in solution or in the final solid pharmaceutical form. If the native structure is not maintained during the lyophilization process, this may be reflected in the unstable final product and, hence, in the product quality, safety, and efficiency [2]. It is also known that proteins can easily fold back during rehydration and exhibit native structures despite partial or complete unfolding in the solid state [3,4]. Nevertheless, the protein may exhibit poor storage stability [5,6]. Therefore, preservation of the native structure in the dried state is crucial for adequate storage stability of proteins [7,8].

Protein drug development is based on stability studies in which critical properties of protein formulations are evaluated using various analytical methods. During these studies, formulations containing proteins are subjected to various forms of stress, including temperature, pH, and freezing, and their properties are monitored over time. The evolution also depends on the choice of excipients, stabilizing agents to protect the protein structure during freezing and drying, and the lyophilization cycle [9]. Usually, several different excipients must be used because different or even opposite protective functions are required. For example, buffers to reduce the pH consequences of concentration changes during freezing [10], solutes that are preferentially excluded from the surface of the drug pharmaceutical to protect against the effect of removal of local water during freezing [11], and also surfactants that can reduce the surface denaturation of proteins during freezing by reducing the ice-water interface [12]. Furthermore, carbohydrates are often added because they can bind to the surface of biological material (the interactions involve hydrogen bonding) and thus protect against drying damage [13,14,15]. Carbohydrates with higher molecular weight have less protective effect than smaller carbohydrates such as sugars, as there are fewer free hydroxyl groups available to interact with protein. Therefore, sugars, such as trehalose, which is known to be particularly effective in this regard [16,17], and others have been used as very effective lyoprotectants. The molar ratio of excipient to protein is also important in stabilizing the protein during lyophilization. In addition, excipients that remain amorphous in the solid state have a better ability to prevent protein aggregation. [18]

Solid-state characterization is currently less established. Usually, analytical methods evaluate critical properties of the protein formulation in solution after reconstitution, which is not necessarily indicative of adequate protein stabilization in the solid phase and therefore cannot guarantee long-term stability of the pharmaceutical form. The less frequent use of solid-phase characterization can also be attributed to the general lack of high-resolution methods. Nevertheless, methods based on structural characterization of proteins in solid form can evaluate both secondary and tertiary structure of proteins in solid-state during formulation development. Thus, only relevant formulations that are able to preserve the native structure of proteins in the solid state can be included in stability studies. The use of such an approach can guide the development of solid-state protein pharmaceuticals and reduce the risk of selecting unsuitable formulations for stability studies as a basis for selecting the final formulation. However, there is a general need for the development of appropriate high resolution analytical techniques for the characterization of proteins in the solid state [19,20].

This review presents the most common analytical methods for the structural characterization of proteins in solids (Figure 1), their basic principles, and a brief discussion with selected examples for each method. For the characterization of protein secondary structure in the solid state, Fourier transform infrared spectroscopy (FTIR), Raman spectroscopy, and near-infrared spectroscopy (NIR) are the most commonly used methods. Solid-state fluorescence and UV–Vis spectroscopy can be used to follow the tertiary structural changes of proteins to some extent. Circular dichroism can be used to study both secondary and tertiary protein structures by measuring the difference in absorbance. Solid-state nuclear magnetic resonance (ssNMR) is used to characterize both structural and dynamic changes. Differential scanning calorimetry (DSC) is useful for characterizing molecular mobility, crystallization kinetics, degree of crystallinity and denaturation, and for determining glass transition temperature (T_g_). Dielectric relaxation spectroscopy (DRS) as a complementary method to DSC can provide insights into protein dynamics, while X-ray diffraction (XRD) analysis can be used to study the powder structure (amorphous/crystalline) of lyophilized proteins. Moreover, recently developed methods based on mass spectrometry, such as solid-state hydrogen-deuterium exchange mass spectrometry (ssHDX-MS), can be used to study the protein structure and conformation in the solid state with high resolution. In addition to structural characterization, monitoring protein aggregation is also very important for the stability, quality, safety, and efficiency of the final drug. Aggregation, along with protein denaturation and surface adsorption, can affect the amount of native protein and, thus, the activity of the drug. This review also presents two regularly used techniques for the analysis of aggregated species: size-exclusion chromatography (SEC) and dynamic light scattering (DLS) [21,22,23].

## 2. Methods for Structural Characterization

### 2.1. Fourier-Transform Infrared Spectroscopy (FTIR)

Examination of protein secondary structure in the solid state immediately after lyophilization can be very useful in predicting long-term storage stability. The preservation of native structure in the dried state can be directly correlated with the prevention of aggregation and protection of activity of labile proteins after rehydration [3]. Infrared spectroscopy is one of the fundamental and most widely used analytical techniques for characterizing protein secondary structures, both in solution and in the solid state. Modern infrared spectrometers are usually Fourier transform infrared (FTIR) spectrometers, since the detector signal is Fourier transformed into the measured spectrum. FTIR spectroscopy can be used to monitor the conformational stability of proteins both prior to lyophilization and after reconstitution, as well as in the lyophilized solids. It is a rapid method, but has some limitations due to water (if present in the sample), which can interfere with the FTIR spectra. Moreover, formulations containing various excipients may be too chemically complex for interpretation. Nevertheless, Fourier transform infrared spectroscopy (FTIR) is a very popular method for studying the loss of native structure or the degree of disruption of secondary structure during freeze-drying of proteins.

Multiple vibrational modes in the mid-infrared range are common for proteins because the amide bonds form the backbone of the protein [24]. The vibrational mode most commonly studied for the analysis of the secondary structure of proteins is the amide I region (1650 cm^−1^) [24,25]. It is directly related to the conformation of the backbone and originates mainly from the C=O stretching vibration with minor contributions from the C–N stretching vibration (Figure 2) [26,27]. In addition, the amide region II (1550 cm^−1^) can also provide useful information about changes in secondary structure derived mainly from in-plane N–H bending [28,29]. Characteristic frequency shifts can be observed depending on the hydrogen-bonding pattern of the amide bonds. This pattern is closely related to the type and amount of structural elements, such as α-helix, β-sheets, and γ-turns [27]. From this information, the secondary structure can usually be deduced. Even though amide I is the region of choice for protein conformational investigations the band arising from water can interfere with protein bands. For protein samples with higher water content, the water-insensitive region amide III (1300 cm^−1^) is preferable for secondary structure analysis. Since proteins are composed of several different secondary structure types, infrared spectra usually consist of many overlapping bands originating from the different structure types. These overlapping bands are difficult to resolve and often result in a single broad amide band. Therefore, mathematical manipulations must be applied to obtain useful structural information. The most commonly used method is spectral derivative (second), which narrows the band and improves the resolution of the signal [30]. There is often some band broadening in the FTIR spectra of freeze-dried proteins. On the other hand, there are also shifts in the band position and intensities of the amide I spectral components due to the physical environmental changes. The shifts can be very significant [7,31]. The formation of protein aggregates, which may be associated with the formation of strongly hydrogen-bonded, antiparallel intermolecular β-sheet structures [32], is closely associated with evidence of the formation of bands around 1620–1625 cm^−1^ and around 1680–1700 cm^−1^ [3,31].

Several studies have been published on structural changes in lyophilized protein formulations using FTIR spectroscopy. Xu et al. predicted protein degradation rates in glassy solid matrices [33], several studies have been published on structural changes in lyophilized protein formulations using FTIR spectroscopy. Xu et al. predicted protein degradation rates in glassy solid matrices [34]. In addition, interpretation of Gaussian deconvolution of amide bands provided insights into estimating the content of secondary structure elements (α-helix and β-sheets) [35]. FTIR was also used to study the effects of mannitol content, for example in the spray-dried antibody formulations [36]. It was used to monitor the effects of sucrose and sorbitol in lyophilizates [37], and for a comparison between different sucrose contents (Figure 3) in selected antibody solids (freeze-dried, foam-dried and spray-dried) [38]. Prestrelski et al. also studied the preservation of native protein structure by disaccharides in the solid state. The infrared spectra of proteins in the pre-lyophilized, lyophilized, and rehydrated states were similar in the presence of disaccharides, except for some formulations containing mannitol [4]. In general, increasing the carbohydrate content in lyophilized formulations resulted in higher intensity of FTIR bands [34]. Samples for FTIR analysis are usually prepared as pellets containing KBr (approximately 5% of the protein), which may have some disadvantages. One of them is aggregation, which can occur during sample separation preparation. Therefore, attenuated total reflectance (ATR) mode has recently emerged as a prominent alternative that is being used more and more frequently [39].

Although infrared spectroscopy is a very useful analytical tool for characterizing freeze-dried proteins (Figure 4), it also has some disadvantages, such as low resolution, semi-quantitative measurement, and poor ability to predict degradation in the solid state. In addition, FTIR can only measure global protein conformations and is unable to monitor local changes that may affect stability. Aggregation, which can occur due to tertiary structural changes, also cannot be detected by infrared spectroscopy, demonstrating the need for a method with better resolution that can measure other than just global secondary structural properties [37,40,41,42,43].

### 2.2. Near-Infrared Spectroscopy (NIR)

Although Fourier transform infrared spectroscopy (FTIR) is the method of choice, near-infrared spectroscopy (NIR) is increasingly used as an analytical tool to study the secondary structure of proteins. It is one of the most valuable non-destructive and non-invasive analytical methods that has found extensive application in the study of pharmaceutical secondary structures [44,45,46,47]. As a non-destructive method, NIR is of particular interest for the study of lyophilized amorphous solid formulations that are prone to moisture absorption [48]. NIR also has several advantages over FTIR and Raman spectroscopy for characterizing protein structure in solids. Unlike FTIR, the instrument does not need to be purged with nitrogen gas, as moisture content tends to show weak interference in NIR spectra. In addition, the experiment time to acquire an NIR spectrum of the analyzed sample is usually no more than 2 min per sample which is quicker than FTIR and much more quicker than Raman [46,49]. Furthermore, there is no sample preparation and no hazardous chemicals are used. In contrast to Raman, samples are preserved during NIR measurements and can be recovered after the analysis (Raman uses laser light that usually damages proteins in samples). While FTIR analyzes require extensive data manipulation to obtain relevant information about protein structure, this is not the case with NIR analyzes [44]. The low absorptivity of water in the NIR range allows for much longer optical path lengths compared to infrared spectroscopy (from 0.5 mm up to 10 mm) [50]. It also allows measurement in reflectance mode, so spectra can be collected directly from the lyophilization vial. NIR can also be used to obtain information on the crystallinity of samples, as well as to determine the residual moisture content in lyophilizates when studying their effect on stability [44,49,51]. As a method, NIR has several advantages over Karl Fischer moisture determination. One of them is undoubtedly that it is a non-destructive analytical tool. Therefore, the same sample can be subjected to a different method for stability studies, thus providing a direct correlation between residual moisture and stability [52,53,54].

Bai et al. investigated the secondary structure of proteins in lyophilized formulations using nondestructive NIR spectroscopy (Figure 5) [51]. In general, different protein secondary structures yield different NIR spectra [55]. Using band narrowing techniques, such as second derivative, we can gain some insight into the correlation between protein absorption bands and different secondary structure elements [56]. For example, the amount of intramolecular hydrogen bonding can predict the conformational stability of a protein in the solid state. A properly folded protein will have many more intramolecular hydrogen bonds, as reflected by a decrease in amide II band frequency [57,58].

In the spectra of lyophilized proteins, bands near 4369 and 4604 cm^−1^ can be associated with an α-helical structure, while β-fold structures are associated with bands at 4323, 4417, and 4525–4535 cm^−1^ [47,56]. It has also been documented that loss of α-helix structure due to exposure to higher temperatures can lead to an increase in β-sheet structure [3,47,56]. NIR has also been shown to be useful for monitoring protein conformation and stability throughout the entire lyophilization process, i.e., from solution to final solid drug product [59,60]. Near-infrared spectroscopy can be a promising alternative to infrared spectroscopy in the biopharmaceutical industry in the development of lyophilized protein pharmaceuticals. It has several advantages over FTIR and Raman spectroscopy, the most important of which are collection of spectra in less than a minute, no sample preparation, and no opening of vials. On the other hand, this method needs further research to be used as a routine method for in-line monitoring of protein secondary structure during lyophilization process. 

### 2.3. Raman Spectroscopy

Raman spectroscopy often complements infrared analysis because it provides information about molecular vibrations and is useful for studying different states of aggregation of biopharmaceutical samples. Similar to infrared spectroscopy, Raman spectroscopy provides sample signature spectra in the fingerprint range. There is also no sample preparation, making it a relatively simple technique compared to some other spectroscopies. Unlike infrared spectroscopy, Raman spectroscopy is based on inelastic scattering. The Raman Effect occurs when a beam of intense radiation, typically from a laser (green, red, etc.), passes through a sample. The protein molecules experience a change in molecular polarizability as they vibrate. The change in polarizability yields information about the peptide backbones and secondary structures of the proteins. Two major advantages of Raman spectroscopy compared to FTIR are the weak scattering and reduced noise due to water. Consequently, in addition to the solid state, proteins can also be studied in their native (aqueous) state [61,62,63]. Typical regions in Raman spectra associated with the components of the secondary structure are the amide I region for the α-helix structure (1600–1700 cm^−1^), the amide III region (1230 to 1340 cm^−1^) and C–C stretching bands (890–1060 cm^−1^) of the protein backbone (Table 1) [64]. The amounts of secondary structure present in the solid protein sample, as measured by Raman spectroscopy in the amide I region, have shown a good correlation with the storage stability of the lyophilizates. However, Raman spectroscopy has also been shown to be a useful tool for screening excipients in formulation design [27,65,66]. Better structure retention has been observed for protein formulations with higher amounts of carbohydrate excipients, such as disaccharides (Figure 6). The significant change in the 1100 cm^−1^ can be attributed to direct correlation of this region with protein unfolding.

Using Raman spectroscopy, Hedoux et al. investigated structural changes caused by lyophilization, and they observed minor changes in terms of protein structure during freezing and rehydration [67]. It was also shown that the spectral changes were mainly caused by vacuum-induced dehydration. Formulations with mannitol as excipient were studied by in-line Raman spectroscopy [68]. The aggregation processes of various proteins were also determined by Raman spectroscopy [69,70,71]. 

Nevertheless, there are also some limitations to the analysis of the secondary structure of proteins by Raman spectroscopy. The laser light can heat the samples locally and damage them. Therefore, the measurements must be performed relatively quickly [66]. Fluorescence noise originating from the sample or impurities can also interfere with the Raman signals.

### 2.4. Solid-State UV–Vis Spectroscopy

UV–Vis spectroscopy allows to study the tertiary structure of proteins and to follow their changes. By using the second derivative of UV spectroscopy, the aromatic residues and their surroundings in proteins can be studied. The main peak of proteins between 240 and 300 nm consists of several spectra that overlap in a final broad peak in the zero-order spectra. These spectra are mainly formed by the absorption of the residues phenylalanine (245–270 nm), tyrosine (265–285 nm), and tryptophan (265–295 nm) [72,73]. Due to the overlap of absorption peaks, UV–Vis spectroscopy has often been limited in the characterization of proteins. By using second derivative analysis as a resolution enhancement technique (as was already done with FTIR), many individual peaks can be derived from the raw spectra with multiple components (Figure 7) [74,75].

The position of the peaks in the second derivative spectra correlates well with the polarity of the microenvironment of the aromatic amino acids and, thus, conformational changes within the protein [76,77]. Therefore, UV–Vis spectroscopy with second derivative processing can be very useful for characterizing the tertiary structure of proteins. Shifts to shorter wavelengths usually indicate increased polarity of the aromatic amino acid microenvironment [78]. On the other hand, the intensity of the peaks can also provide useful structural information. Derivative intensities can be used to calculate amino acid content [72,79], and the so-called a/b ratio is used to evaluate the exposure of tyrosine to bulk solvent [76]. In recent years, the sensitivity of UV–Vis spectroscopy has also greatly increased and now second-derivative negative peaks can be measured with a resolution of up to 0.01 nm, providing a very sensitive tool to study protein conformational changes [78,80,81]. Second derivative UV–Vis spectra can also provide us with information about the perturbations of the tertiary structure of proteins as a function of various conditions, such as pH (Figure 8), temperature, ionic strength, and other factors [81,82,83]. Temperature dependence of second-derivative peak positions has been widely used as a tool to study the thermal unfolding of proteins. Shifts to lower wavelengths at elevated temperature usually indicate increased exposure of the aromatic side chains to the solvent. Another example is the study of aggregation behavior as a function of pH and temperature [84].

### 2.5. Solid-State Fluorescence Spectroscopy

Despite the fact that altered secondary structure in proteins directly implies loss of tertiary structure, preserved secondary structure does not confirm retention of tertiary structure. Therefore, it is crucial to systematically study the tertiary structure of proteins as well. Recent literature has shown that solid-state fluorescence spectroscopy can be a useful tool to study the tertiary structure of proteins in solid dosage forms, and that tertiary structure correlates with long-term stability [85,86,87,88,89].

In solid-state fluorescence, the molecules of proteins are excited to a wavelength corresponding to their excitation maximum. Light is then emitted during the transition from the excited state back to the ground state. Aromatic amino acids present in proteins provide intrinsic fluorescence signals that can be measured, and they provide useful information about tertiary structure. The most common is tryptophan fluorescence, which is often used to check the integrity of the tertiary structure of proteins [90,91]. Solid-state fluorescence has been used to study the unfolding of proteins in different environments such as pH, temperature, and buffer excipients to provide direct information on the degradation of freeze-dried proteins (Figure 9) [85,89]. In another study, a monoclonal antibody freeze-dried with sucrose as an excipient and stored at a higher temperature showed decreased fluorescence intensity as well as quite large aggregation [86]. Proteins can also be derivatized with fluorescent probes to detect protein aggregates. In this case, the sensitivity of solid-state fluorescence is increased at wavelengths characteristic of the probes [92].

Strong background scattering of lyophilized protein powders with high optical density may prove to limit solid-state fluorescence. This problem can usually be avoided by analyzing samples in front-face mode. Another disadvantage of solid-state fluorescence is its sensitivity, which depends on the position of the aromatic amino acids in the protein tertiary structure. This can be solved by co-lyophilization of proteins with chromophores [93].

### 2.6. Circular Dichroism (CD)

Circular dichroism (CD) measures the difference in protein absorbance for left- and right-handed circularly polarized light. The difference in absorbance is due to the interaction of protein chromophores with the chiral environment [94]. Typical applications with CD analysis include estimating the content of protein structures (secondary and tertiary) and detecting conformational changes in a protein due to changes in the environment (pH, excipients, addition of denaturants, and temperature) [83,95,96,97]. CD can measure in both the near-UV CD region (310–255 nm) and far-UV CD region (below 250 nm) of the spectra. Tryptophan, tyrosine, phenylalanine, and cystine contribute to the signals in the near-UV CD spectra. The signals provide direct information about the tertiary structures of the proteins. On the other hand, peptide-amide bonds contribute to the signals in the far-UV CD spectra. These signals reflect the secondary structure of the protein, i.e., α-helixes, β-sheets and γ-turns [98]. 

Liu et al. investigated the forced oxidation of proteins by hydrogen peroxide. The data obtained clearly indicate disruptions in the tertiary structures. On the other hand, most of the secondary structure was retained [99]. Harn et al. also studied proteins with CD, but as a function of temperature. Increasing the temperature at rather small intervals carried out thermal studies. The changes in the relative content of the β-sheet structure of the protein were observed. It was shown that one antibody had better stabilization than the other at the same protein concentration [100]. CD has also been used as a tool to monitor the aggregation of antibodies to assess the stability of therapeutic proteins during process development, formulation development and product characterization [101]. Similarly, Vermeer and Norde used CD in combination with other techniques, such as DSC, to assess the thermal stability of immunoglobulin, its unfolding and aggregation [102]. Ramachander et al. also used CD as an additional method to solid-state fluorescence spectroscopy to analyze changes in the secondary and tertiary structure of the reconstituted protein (Figure 10) [85]. 

Circular dichroism experiments are typically measured in solution, i.e., water or buffer with a pH from 2 to 12. In recent years, there has also been the option for measurements in solid-state mode, which could be quite useful for studying lyophilized protein pharmaceuticals [103,104]. Further, there are several interesting works dealing with solid-state CD, such as the study by Kawamura et al. on L- and D-serine in combination with solid-state NMR and density functional theory (DFT) [105]. Moreover, Bak et al. studied the effect of high pressure treatment on soluble proteins [106].

### 2.7. Solid-State Nuclear Magnetic Resonance Spectroscopy (ssNMR)

Solid-state nuclear magnetic resonance spectroscopy is a powerful method for structural characterization as well as for studying the dynamics of lyophilized protein pharmaceuticals [107,108]. With ssNMR, proteins can be analyzed with atomic-level resolution. However, the random orientations of proteins in solid form limit the use of ssNMR to provide high-resolution structural information. ^1^H (proton) and ^13^C (carbon) are the two most commonly used nuclei in NMR spectroscopy. Normally, large magnetic fields are preferred as they offer increased sensitivity.

With ssNMR, relaxation measurements can be performed on dried protein formulations, with very good correlations between relaxation times and pharmaceutical stability in solid formulations, even in the long-term range [109,110,111,112,113,114]. In addition to chemical analysis, ssNMR determines molecular mobilities over a wide range of time scales for β-relaxation: via spin-lattice relaxation times (T_1_) on a picosecond-nanosecond time scale and spin-lattice relaxation times in a rotating frame (T_1ρ_) on a microsecond-millisecond time scale. Whereas T_1p_ is measured in a rotating frame, the T_1_ is measured in a laboratory frame [115,116]. ssNMR spectroscopy can also be used to study the interactions between proteins and excipients by measuring the differences in chemical shifts in different formulations. A nice example can be found in the work of Lam et al. in which the stability of lyophilized formulations, containing lactose and trehalose as excipients, was studied (Figure 11) [116]. The addition of sugars reduced the relaxation rates, whereas the presence of moisture increased them. Although changes in T_1p_ may be random and rather small, they usually provide information about the long-term stability of proteins in the solid state, since relaxation times are directly related to the frequencies of molecular motion. In general, lower values for the relaxation times represent greater long-term stability of the measured sample. In addition, Separovic et al. found strong correlations of T_1_ relaxation times with changes in protein aggregation and activity [117]. Yoshioka’s group also extensively studied the molecular dynamics of proteins using ssNMR spectroscopy. They showed a good correlation of relaxation times with aggregation rates of freeze-dried proteins [118,119,120]. Furthermore, Tian et al. have shown that the stability provided by arginine in freeze-dried antibody products is due to non-covalent interactions between the arginine side chain and the protein [31]. The study of protein relaxation times using ssNMR can also be used to predict storage stability in the solid state, as there is a very good correlation between ssNMR data and long-term storage stability [89].

### 2.8. Differential Scanning Calorimetry (DSC)

Differential scanning calorimetry (DSC) is one of the most common techniques for characterization of freeze-dried protein drugs [121]. It is commonly used as a complementary method to spectroscopic and chromatographic characterization of proteins in the solid state [122]. The method consists of heating and cooling the sample together with a reference, measuring the difference in the amount of heat required to raise the temperature of the sample and reference. The difference is measured as a function of temperature, maintaining similar temperatures for the sample and reference [123]. DSC is a widely used technique for studying the unfolding of protein secondary structures and for characterizing the conformational stability of proteins under various conditions. DSC can be used in both solid and liquid states. Nano differential scanning calorimetry (nanoDSC) is used to analyze liquid samples of protein formulations. Lyophilized proteins are usually dissolved in buffers with pH around neutral, but water can also be used. From the heat capacity curve, the melting temperature (T_m_), as well as the calorimetric enthalpy (ΔH_m_) and entropy (ΔS_m_) of the unfolding process can be accessed. For example, nanoDSC has been used to determine the melting temperature of proteins with FTIR spectroscopy [124]. On the other hand, DSC is more commonly used to study solid samples of protein formulations. While the unfolding of proteins produces endothermic peaks, their aggregation is shown to be an exothermic event. Moreover, in the study by Pikal et al. aggregation could be correlated with the loss of secondary structure and a decrease in the area under the denaturation endotherm. This process was reversible with trehalose as an excipient in the formulations [122]. By comparing the melting temperatures (T_m_) of different protein lyophilizates, insights can be gained into their different secondary structures. DSC can also be used to optimize lyophilization parameters, ultimately resulting in a better appearance of the lyophilized protein cake. For example, Han et al. investigated the effect of sucrose and mannitol. It was found that the addition of sucrose to the formulations resulted in an upward shift in melting points, thus providing protection, while the addition of mannitol did not show the same benefit [125]. Ihnat et al. also investigated the effects of various excipients on the thermal stability of lyophilized protein samples [126].

Interestingly, protein stability does not always correlate well with glass transition temperature (T_g_) values [127]. On the other hand, formulations stored above their T_g_ are generally less stable than those stored below. While storage below the T_g_ is necessary, it is not always sufficient to ensure stability. Therefore, when evaluating protein stability in lyophilized formulations, DSC characterization should be accompanied by other analytical methods [6,128,129]. In addition, modulated DSC (mDSC) can resolve total heat flow into thermodynamic (reversal) heat flow and kinetic (non-reversal) heat flow. For example, the enthalpy relaxation endotherm (kinetic) can be separated from the glass transition event (thermodynamic) [130]. Therefore, mDSC is particularly useful for detecting T_g_ present with other overlapping thermal events (glass transition) [131]. Therefore, mDSC is particularly useful for detecting T_g_ present with other overlapping thermal events (glass transition) [132], to correlate eutectic temperature (lowest possible melting temperature over all of the mixing ratios for the involved component species) events leading to cake collapse [133] and to study the effects of annealing on the thermal properties of frozen sucrose solutions [134,135,136]. It was also used as an additional method to ssHDX-MS for the analysis of the conformation of lyophilized IgG1 (Table 2) [137].

### 2.9. Dielectric Relaxation Spectroscopy (DRS)

Dielectric relaxation spectroscopy (DRS) is a thermoelectric technique for studying changes in the conformational dynamics of proteins. It is a useful analytical tool that can complement calorimetry. It is a non-invasive method that characterizes protein motions over a frequency range from 10^–5^ to 10^11^ Hz. DRS determines the time dependence and extent of electrical polarization processes by measuring the speed and extent of polarizability of a material placed in a weak and sinusoidal oscillating electric field [138,139,140]. The wide frequency range allows the study of slow and hindered macromolecular oscillations, restricted charge transfer processes, as well as relatively fast reorientations of small molecules or side-chain groups. Thus, the method can distinguish relatively well between groups involved in global and local dynamics. This is very important because protein dynamics involves many different types of motion, both local and more global, including transitions between multiple conformational substates. More specifically, DRS reflects the mobility of molecular dipoles and is able to directly capture the time scale of molecular motions. High-frequency studies, for example, refer to specific dipole reorientation polarizations, and are directly linked to the microscopic structure of the solid sample. Thus, the dipoles act as molecular probes that can provide information about the structural properties of the sample. In addition, DRS can provide information on protein structure, primary and secondary molecular motions (dynamics), water content, and its state (bound or free). The dielectric analysis of water in pharmaceutical systems is typically done in the high-frequency region. [134,141,142]. 

Different lyophilized protein formulations with variable amounts amount of sugar and protein drug were analyzed by E. Mozine et al. using DRS. Significant differences in dielectric relaxation times and activation energies were observed [134,138,139]. In addition, DRS can also be used to study the correlation between temperature and relaxation kinetics. Thus, the dielectric measurements of samples are performed over a range of temperatures. Furthermore, the molecular mobility of freeze-dried protein pharmaceuticals under the Tg was measured in relation to stability during storage. In general, a longer relaxation time for sucrose and trehalose formulations was associated with better stability [22]. In addition, studies of the hydration effect on protein dynamics in the lyophilizates were performed using DRS analysis (Figure 12) [140,143]. Nevertheless, the correct assignment of the DRS spectra of the hydrated solid-state pharmaceuticals can be challenging because the relaxation kinetics of water and protein drug can overlap. Therefore, the moisture present in the samples generally poses a problem and additional studies may be required to use DRS as an analytical tool in the development of drug formulations.

### 2.10. X-ray Diffraction (XRD)

Diffraction analysis based on X-ray light is a widely used analytical technique to study the physical state of materials, which has been used in drug research and development for many years. It is a method of choice for three-dimensional structural analysis of proteins in the solid state. The technique provides information on powder structure (amorphous/crystalline) and internal surface structure for better understanding of structure–function relationship of proteins and their interaction with various factors in their environment. Changes in the crystalline form of protein pharmaceuticals may be reflected in changes in solubility, which has implications for bioavailability. In addition, changes in the formulation matrix can affect the stability of lyophilized proteins [144,145,146]. In X-ray powder diffraction (XRD), an X-ray beam is incident on a single crystal and scattered at different angles. The intensities and angles at which the X-rays are scattered allow the electron density in the crystal to be determined, which in turn leads to the derivation of a three-dimensional structure of the crystal. Thus, XRD is ideally suited to monitor the evolution of crystallinity as well as changes in the crystalline forms of lyophilized protein pharmaceuticals.

Crystallization of the components in the lyophilized proteins formulations can be induced, for example, by lowering the temperature during the lyophilization cycle [147,148]. Crystallization of the formulations may result in individual components not having the same physical form as present in the starting materials. For example, an excipient may crystallize or remain amorphous. When the excipient crystallizes, there are several possibilities, such as crystallizing a polymorphic form or a hydrate of the starting form. These forms have already been observed in various formulations, demonstrating the need to also examine the lyophilized protein pharmaceuticals also by powder X-ray diffraction. Such changes in the form of excipient may lead to degradation of proteins as active species in the formulations. There are several studies on the behavior of excipient mannitol with XRD as the analytical technique used. Mannitol is a very good example, as it can exist in at least three polymorphic forms, an amorphous form and a hemihydrate form (Figure 13) [149,150,151,152,153,154].

In a recent study by Norrman et al., the structure in insulin formulations was investigated. In these formulations, protamine was co-crystallized with insulin. XRD analysis showed that protamine acts as a charge balancer and does not specifically bind to insulin [155]. Izutsu et al. in which the stabilizing effect of amphiphilic excipients and sugars was investigated by XRD analysis published another interesting study. It was shown that various amphiphilic excipients, such as polyethylene glycol (PEG), act as good stabilizers for lyophilized proteins when dispersed in a sugar-dominated matrix [17].

### 2.11. Solid-State Hydrogen-Deuterium Exchange Mass Spectrometry (ssHDX-MS)

Hydrogen/deuterium exchange mass spectrometry (HDX-MS) has recently emerged as a very useful and efficient method for studying protein conformation and interactions with excipients in the solid state, as well as for predicting physical stability. As a high-resolution method for analysis in solution, HDX-MS has been used for many decades to study protein structure, conformation, stability, solvent exposure and dynamics [156,157,158,159,160], as well as protein folding and ligand binding [161]. It measures the kinetics of deuterium uptake in both the intact protein and its peptic fragments [162]. HDX-MS as a technique consists of exposure of the protein to D_2_O, whereupon the rate as well as extent of deuterium incorporation is recorded as a change in mass (m/z) [163,164]. Similarly, solid-state hydrogen-deuterium exchange mass spectrometry (ssHXD- MS) can be used to measure protein structure, interactions with excipients, and to study the conformation of proteins in lyophilized powders. Results are usually obtained with relatively high resolution and the method is characterized by a very good correlation between deuterium exchange and physical stability during storage [43,165]. In practice, freeze-dried protein samples are exposed to D_2_O in the vapor phase at controlled relative humidity, vapor pressure, and temperature for varying lengths of time. This is often done in a desiccator containing deuterium oxide. Samples are then rapidly reconstituted under quenched conditions (usually in acidic buffer with a pH around 2.5) and subjected to liquid chromatography-mass spectrometry (LC/MS) analysis with or without proteolytic digestion. If digestion is performed prior to analysis of the samples using LC/MS, information on deuterium uptake can be obtained with peptide-level resolution. Such an approach is often used with HDX-MS in solution [166,167,168]. Then, the extent of deuterium uptake by the protein and its peptide fragments is measured as a function of time using mass spectrometry [20,169]. Indeed, hydrogen atoms can be exchanged for deuterium upon exposure to D2O. On the other hand, they can also be exchanged again, so that only the hydrogen atoms on the amide backbone can be analyzed by mass spectrometry. Such analysis can provide insight into the intra- and intermolecular hydrogen bonding interactions in the formulation matrix. These interactions can strongly influence the physical stability of the protein drug [170].

Although HDX-MS is a relatively new analytical technique, there are already several examples of its application to the study of protein-based pharmaceuticals in solid form. French et al. studied spray-dried powders with HDX and FTIR, taking advantage of isotopic shifts in the amide bands [171]. Desai et al. used HDX with proton NMR analysis to study the protein structure of pancreatic trypsin inhibitor in the lyophilized state [172]. ssHDX was also used to study hydration in lyophilized myoglobin. The extent of HDX in solids is related to hydration of the exchangeable amide groups and protein conformation and dynamics. By using pepsin digestion, mapping with resolution at the peptide level can also be achieved [173]. In addition, the kinetics of HDX were also investigated, suggesting that it is useful for detecting non-native species in protein formulations that cannot be detected by other methods, such as FTIR [174]. Moorthy et al. used ssHDX to predict protein aggregation of myoglobin in lyophilized formulations. They showed a better correlation with the extent of aggregation for ssHDX compared to FTIR band intensity [43]. Similar results were obtained by examining the conformation and aggregation of a monoclonal antibody IgG1 (immunoglobulin G1) (Figure 14). Interestingly, protein aggregation decreased when histidine was added to formulations containing sucrose. However, these results did not correlate with structural or conformational changes observed by FTIR or HDX-MS [137]. Wilson et al. used ssHDX- MS to investigate the effects of processing conditions and excipients on protein structure and physical stability. The results obtained show, among other things, how this method can be used as a tool not only for predicting stability but also for developing more robust protein formulations [175]. 

Solid-state HXD-MS as a characterization tool can be very useful for solid-state protein stability studies, providing information at higher resolution than other conventional methods [176]. While conventional analytical tools can detect differences between processes and formulations, there is no good correlation with the physical stability of proteins. On the other hand, ssHDX-MS shows a greater correlation to physical stability. Usually, a more stable sample has a lower maximum deuterium retention (D_max_) and peak area. Furthermore, ssHDX can also be used to study population heterogeneity within a protein formulation. For example, spray-dried formulations have shown better heterogeneity compared to the corresponding lyophilized samples. Although ssHDX- MS has proven to be a useful tool for predicting the stability of solid pharmaceutical proteins, it can also be used to develop more robust protein formulations.

## 3. Methods for Aggregation Studies

### 3.1. Size-Exclusion Chromatography (SEC)

As a loss of higher order structure, protein denaturation can occur and lead to aggregation or chemical instability. When the protein is denatured (partially or completely), more hydrophobic parts are exposed, as well as greater flexibility of the whole molecule [177,178]. However, aggregates and denatured protein species may still have a considerable amount of secondary structure and are therefore indistinguishable from native species using techniques such as FTIR spectroscopy [95]. This could be due to the lack of sensitivity of FTIR spectroscopy, combined with secondary structural changes in only small parts of the protein leading to aggregation [36]. Because spectroscopic techniques provide only general information about the overall secondary structure of the sample, aggregation could also be caused by a change in the secondary structure of only a small population of molecules that may not be detected [39]. Therefore, significant parts of the secondary structure are retained upon aggregation and only changes in tertiary structure may occur. Finally, proteins in lyophilized pharmaceuticals can aggregate without any detectable change in secondary structure [179].

Size-exclusion chromatography (SEC) has emerged as the method of choice for the detection and study of protein aggregation, particularly for drug discovery applications [176,180,181]. SEC can separate protein species based on their hydrodynamic size under native conditions. SEC columns can be used in both high-performance liquid chromatography (HPLC; formerly called high-pressure liquid chromatography) and ultra-high-performance liquid chromatography (UHPLC). Although primarily intended for the detection of soluble aggregates, SEC can also be used for the analysis of truncated species (fragments) in a protein sample [182]. 

SEC columns are usually silica-based with high recovery and resolution. Proteins from 10 kD up to 150 kD in size (protein pharmaceuticals) have been successfully separated and analyzed using such columns. For hydrophobic proteins, which tend to have poor recovery, polymer-based columns can be used, but at the expense of resolution compared to their silica-based counterparts. Mobile phases for SEC analysis of lyophilized proteins are typically phosphate-based buffers containing 100 to 500 mM sodium chloride, at a pH near neutrality. Sodium chloride can be substituted with sodium sulfate or even sodium perchlorate [183]. To prevent tailing or poor recovery for proteins that may interact too strongly with silica columns, arginine was used to minimize adsorption [184].

Using SEC analysis, both high molecular weight (HMW) species belonging to aggregates and low molecular weight (LMW) species can be detected and characterized. The LMW species represent degradation (denaturation) products, e.g., from backbone hydrolysis or other fragmentation [89]. 

A nice example of a SEC study in the context of formulation development is the work by Wang et al. on the influence of sucrose content on the storage stability of proteins, where the correlation between aggregation, protein structure, and molecular mobility was investigated [185]. Another article worth mentioning is the review paper by Goyon et al. on modern SEC, where the current possibilities for characterization of protein pharmaceuticals are discussed. Some of the SEC chromatograms from this study are shown in Figure 15 [186]. Furthermore, Andya et al. used SEC in combination with some other methods such as DSC, FTIR, and CD to investigate the mechanisms of aggregation and stabilization by carbohydrate excipients [32]. Last but not least, SEC has also been used together with ssHDX- MS to study the effect of drying method and excipient on structure and stability of protein solids (Figure 16) [175].

### 3.2. Dynamic Light Scattering (DLS)

As a non-destructive complementary method to Size-exclusion chromatography (SEC), dynamic light scattering (DLS) can be used to detect protein aggregates that cannot be detected by SEC analyzes. A typical DLS study data results can be seen in Figure 17 [187]. DLS is one of the size-based analyzes used to determine the size distribution of particles. Its principle is based on the fact that the intensity of light scattered from a sample is proportional to the particle size and its concentration. DLS can be used to measure particles in the diameter range from 1 nm to 5 μm, and the results can be reported in a volume-based distribution [181,188]. Because it is a qualitative screening technique for characterizing protein solutions, it can be used to distinguish a homogeneous from an aggregated sample. Although it has good sensitivity for detecting large aggregates, it is difficult to perform quantification analysis [181,189]. Its resolution is also limited [190]. However, DLS is generally used as an analytical tool for comparing different samples rather than for absolute measurements. Therefore, different lyophilized protein formulations are usually analyzed and compared [191].

DLS has been used as a standard method for determining protein size analysis and aggregation patterns [192]. Furthermore, DLS was also used for characterization in the PEGylated erythropoietin (EPO) study [193]. In addition, DLS was used in monitoring the aggregation of carbonic anhydrase [194,195], and β-amyloid peptide [196,197]. Hawe and Friess studied the correlation between pH and aggregation and showed that the percentage of aggregated species increased with higher pH values [198]. DLS was also used to study the effects of excipients on aggregation [199] and to study the effects of pH and buffer concentration on the thermal stability of etanercept, a biopharmaceutical used to treat autoimmune diseases [200]. In addition, Sukumar et al. used dynamic light scattering to check the opalescence of some IgG1 antibody samples for possible aggregation at high concentrations [201].

## 4. Method Summary

Table 3 gives an overview of the methods described above with some of the “must know” information.

Table 4 presents a comparison of different methods used to study same or similar samples of lyophilized formulations. Powders containing one of proteins (myoglobin, bovine serum albumin, lysozyme, β-lactoglobulin) were formulated with sucrose, trehalose, or mannitol and dried using lyophilization or spray-drying. The samples were analyzed by FTIR, fluorescence, XRD, DSC, and ssHDX-MS. Protein instability was determined by loss of monomeric peak using SEC [175].

## 5. Conclusions

Since the increased use of protein-based biopharmaceuticals in the form of lyophilized solid powders, there is a high demand for good analytical methods with high resolution to characterize these drugs. Preservation of the native structures of the proteins is critical for the development of a safe, efficient, and high quality drug. Although most analytical methods evaluate the properties of proteins in solution, an increasing number of methods for solid-state characterization of protein pharmaceuticals have been used. These methods can characterize and monitor changes in the secondary and tertiary structures of proteins with relatively good accuracy. FTIR and Raman spectroscopy can be routinely used to evaluate secondary structure in biopharmaceutical products and their changes. FTIR has the advantage of providing sample spectra quickly and is a nondestructive method that has simple and not expensive equipment. Raman analysis, on the other hand, takes a little more time to get a correct measurement, but is a complementary method to FTIR. NIR spectroscopy is a non-invasive and non-destructive technique that can monitor the secondary structure of proteins, and serves as an in-line tool for tracking structural changes throughout the whole lyophilization process. Similar to FTIR, NIR also has equipment that is not expensive and easy to set up. Samples can be analyzed and recovered in just a few minutes. In addition, no inert gas purge is required. On the other hand, FTIR, NIR, and Raman do not correlate as well with storage stability and analysis can only be done on a global scale. Solid-state fluorescence and UV–Vis spectroscopy can be used to derive information about the tertiary structure of proteins. These two methods are similar to FTIR and NIR, delivering spectra in short time and preserving the sample during the analysis. Moreover, here the results usually do not correlate well with aggregates formation. CD can provide insight into both secondary and tertiary structural changes. The advantage of this method is that one can measure the degree of structure both in the solid and in the solution. The disadvantage of the method is that in some cases the changes in the spectra are rather small and concentrated samples must be used. XRD analysis is the method of choice to obtain information on powder structure (amorphous/crystalline). While the crystallinity analyses are very good, the samples here are damaged by X-ray irradiation and, therefore, cannot be recovered. In addition, the measurements are made on a global scale. Protein dynamics is generally studied using ssNMR, DSC, and DRS. The information from protein dynamics has shown very good correlation with their stability in the solid state. These three methods are sensitive, but the sample cannot be recovered. This can be a particular problem with ssNMR, as a relatively large amount of sample is required. All three methods require a certain amount of time to produce results (the ssNMR experiment may run overnight, for example). ssHDX-MS has emerged as a relatively new tool for studying protein structure and dynamics, showing better correlation with physical stability than some of the conventional characterization methods mentioned above (very good correlation with aggregates formation). Another advantage is the ability to analyze samples on a global and local scale. On the other hand, the method requires relatively expensive equipment and the measurements are not as fast as other routinely used techniques. Moreover, ssHDX- MS can also reveal population heterogeneity within a protein formulation. Finally, for the study of aggregation phenomena, which is of great importance for successful drug development, SEC and DLS have been found to be the most reliable and widely used analytical techniques. SEC is the method of choice here because it can give good and reliable results with good resolution, whereas DLS sometimes cannot distinguish some particle types. On the other hand, DLS can detect some particles that are not observed with SEC. Therefore, the two methods complement each other, and it is advisable to analyze the samples with both.

## Figures and Tables

**Figure 1 pharmaceutics-13-00534-f001:**
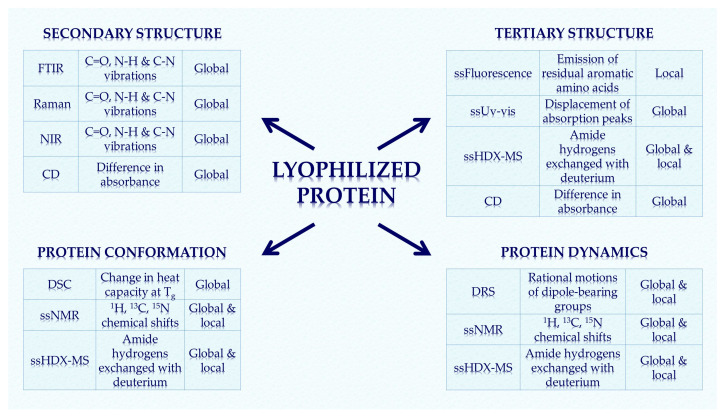
The most common analytical techniques for the structural characterization of proteins in solid pharmaceutical forms are presented with corresponding type of measurements. Changes in secondary/tertiary structure and conformation can be studied on a global and local scale. Protein dynamics can also be traced using some of the above methods. FTIR—fourier transform infrared; NIR—near-infrared; CD—circular dichroism; ss—solid-state, HDX-MS—hydrogen-deuterium exchange mass spectrometry; DSC—differential scanning calorimetry; NMR—nuclear magnetic resonance; DRS—dielectric relaxation spectroscopy.

**Figure 2 pharmaceutics-13-00534-f002:**
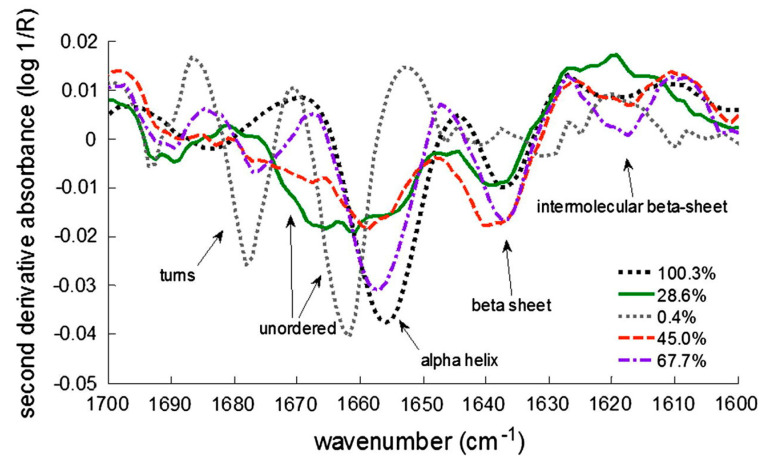
Second derivative FTIR (fourier transform infrared) spectra of lyophilized lactate dehydrogenase (LDH). Different types of secondary structures are shown, as well as different % of remaining LDH activity after reconstitution. Reproduced with permission from [24], Elsevier, 2013.

**Figure 3 pharmaceutics-13-00534-f003:**
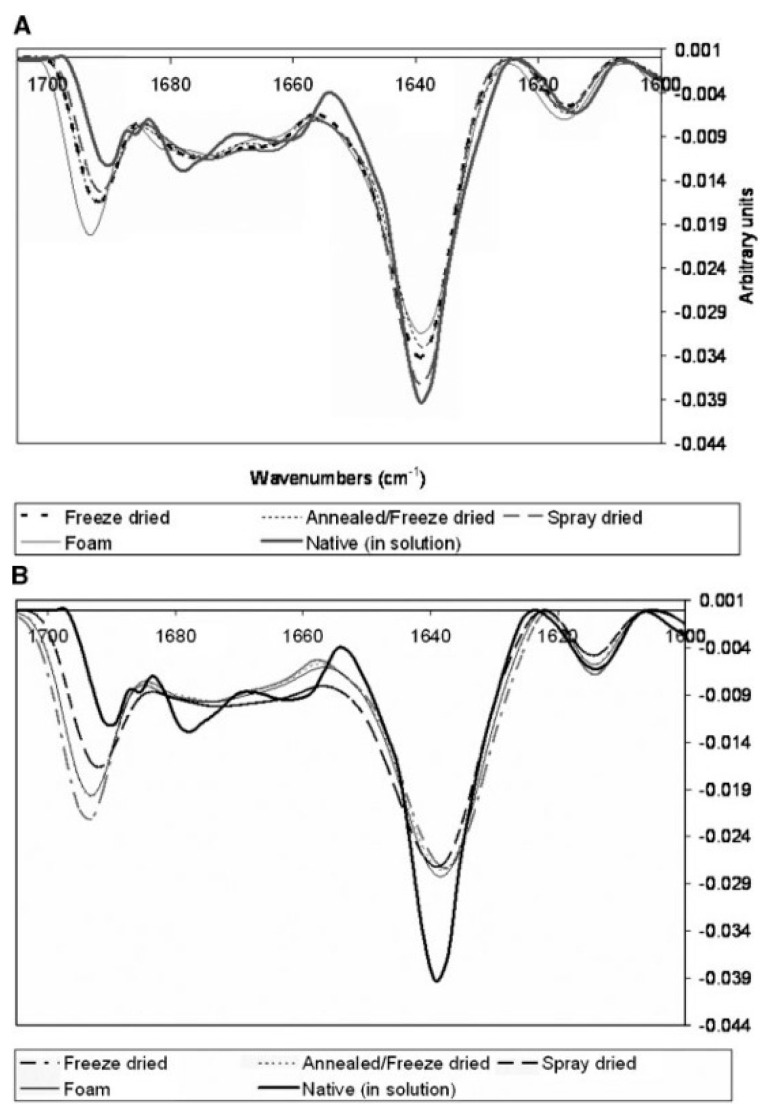
Second derivative FTIR spectra of immunoglobulin G1 (IgG1) in solid form in two different formulations with sucrose—sucrose:IgG1 = 4:1 (**A**) and sucrose:IgG1 = 1:4 (**B**). Formulations with higher sucrose content show better stability and thus preservation of the native secondary structure. Reproduced with permission from [33], Wiley, 2007.

**Figure 4 pharmaceutics-13-00534-f004:**
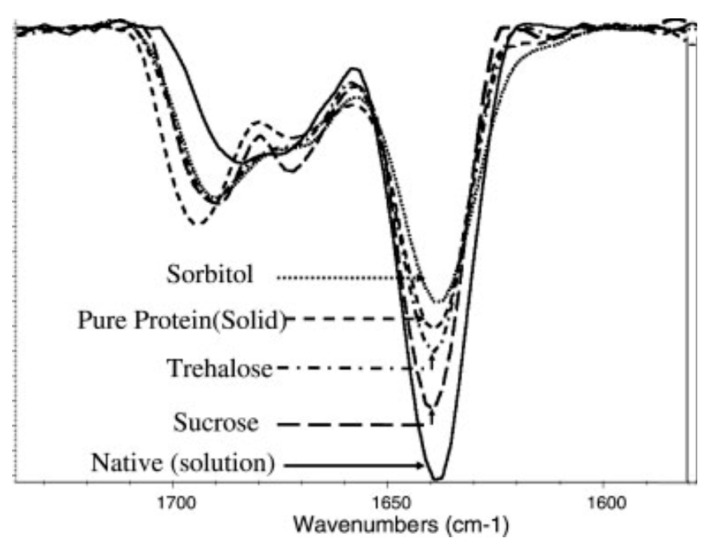
Second derivative FTIR spectra of lyophilized antibody formulations with different excipients used. The ratio between the sugars and the antibody is 1:1. Native structure spectra were recorded with pure antibody in 5 mM phosphate buffer at pH 7. Reproduced with permission from [41], Elsevier, 2005.

**Figure 5 pharmaceutics-13-00534-f005:**
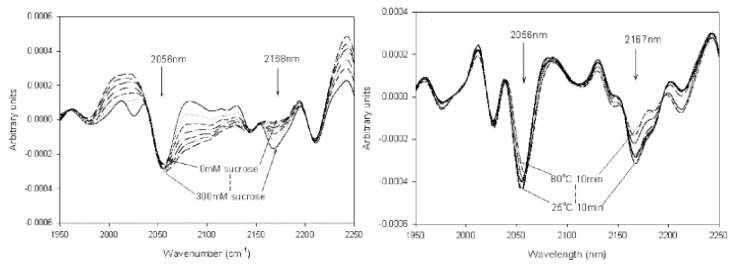
NIR (near-infrared) spectra as second derivatives of structural studies on lyophilized cytochrome c. The influence of sucrose content on the preservation of secondary structure is shown in the left spectra, while the influence of temperature on protein stability can be seen in the right ones. Reproduced with permission from [51], Wiley, 2005.

**Figure 6 pharmaceutics-13-00534-f006:**
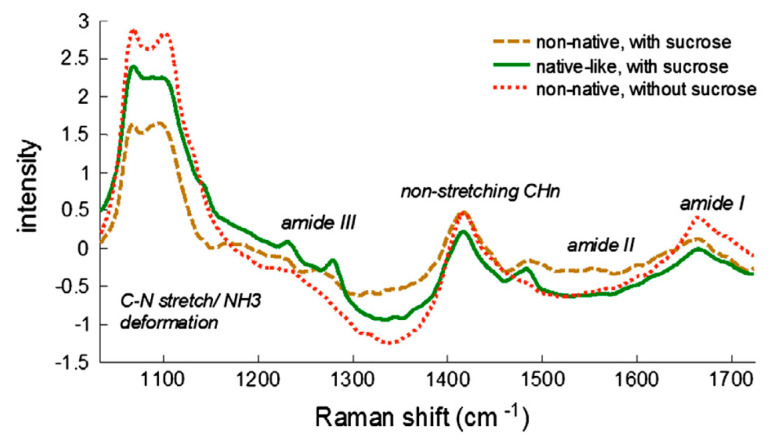
Raman spectra of native-like and denatured (non-native) freeze-dried LDH. Sample concentration 9 mg/mL, 6.5 mg of material used. Reproduced with permission from [24], Elsevier, 2013.

**Figure 7 pharmaceutics-13-00534-f007:**
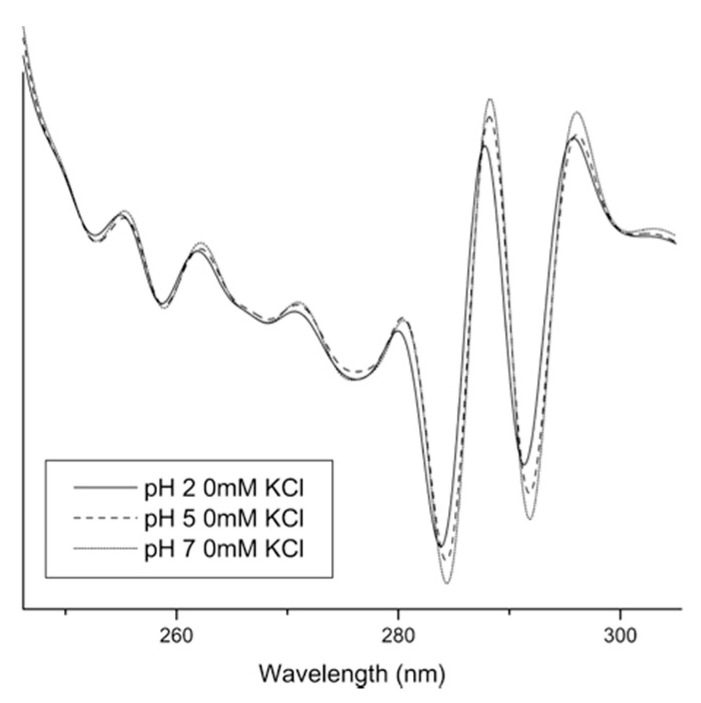
Second derivative absorbance spectra for UV–Vis analysis of IgG at three different pH values without KCl. The largest shifts were observed for the two peaks at approximately 284 and 292 nm. Reproduced with permission from [76], Elsevier, 2008.

**Figure 8 pharmaceutics-13-00534-f008:**
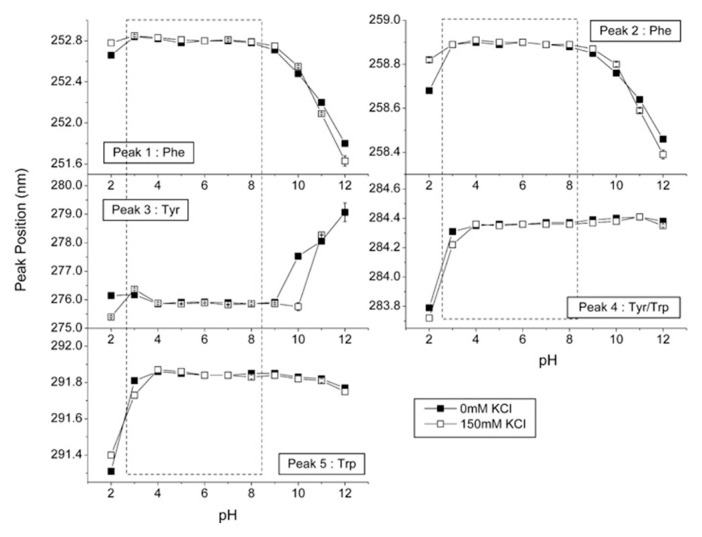
Second derivative UV–Vis spectra for IgG. The effect of different pH is presented for five characteristic peaks. Reproduced with permission from [76], Elsevier, 2008.

**Figure 9 pharmaceutics-13-00534-f009:**
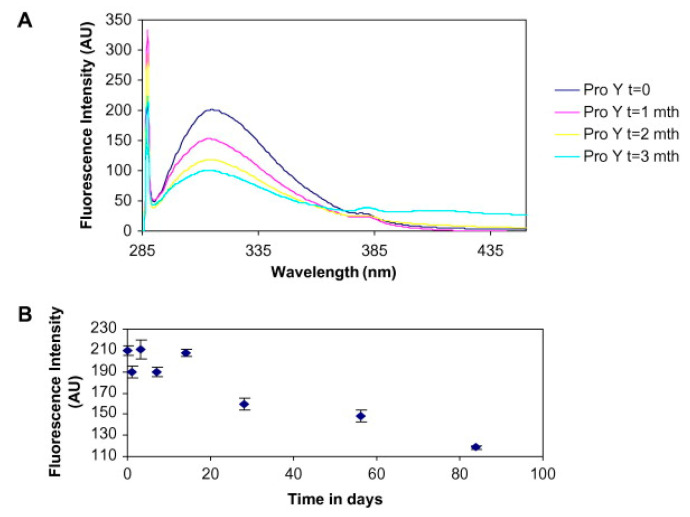
Solid-state (ss) fluorescence study of temperature stress effect on lyophilized protein stability. Subfigure (**A**) presents the fluorescence spectra at various time points during incubation at 60 °C. The decrease in intensity can be clearly observed. Further, the subfigure (**B**) presents the absolute fluorescence intensity versus time Reproduced with permission from [85], Elsevier, 2008.

**Figure 10 pharmaceutics-13-00534-f010:**
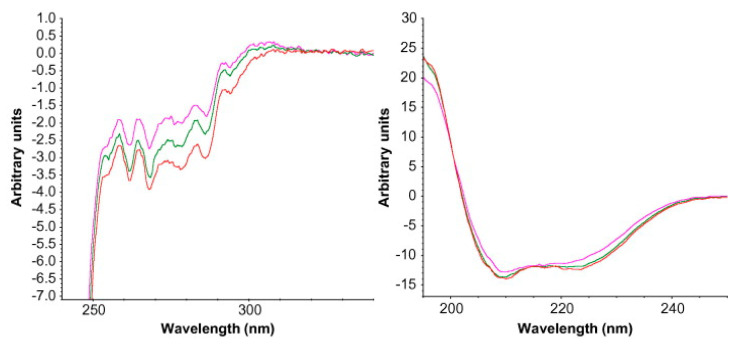
Circular dichroism (CD) spectra for reconstituted lyophilized protein incubated at two different temperatures. Changes in tertiary structure are seen in the near-UV CD spectra on the left, while changes in secondary structure are shown in the far-UV CD spectra on the right. Red spectra—lyophilized protein (control), green spectra—incubation at 37 °C, pink spectra—incubation at 60 °C. Reproduced with permission from [85], Elsevier, 2008.

**Figure 11 pharmaceutics-13-00534-f011:**
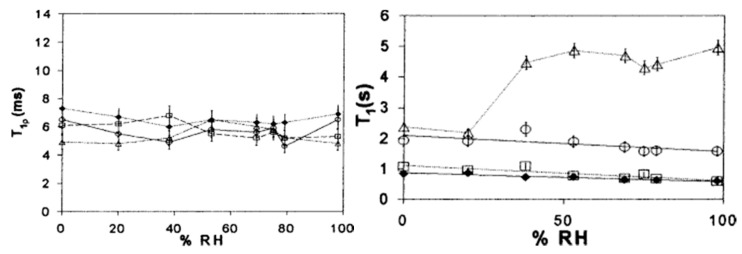
ssNMR study of several lyophilized protein formulations containing different amounts of lysozyme and trehalose. On the left average T_1ρ_ is plotted against relative humidity (% RH), whereas on the right average T_1_ is plotted against % RH. 
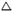
—trehalose; 
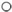
—80% trehalose, 20% lysozyme; □—20% trehalose, 80% lysozyme; 
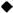
—lysozyme. Reproduced with permission from [116], Elsevier, 2002.

**Figure 12 pharmaceutics-13-00534-f012:**
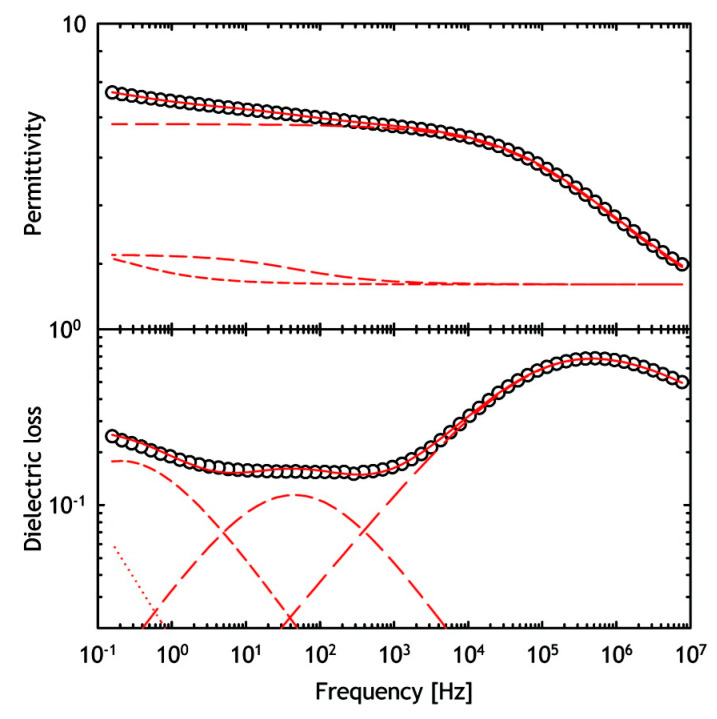
Typical dielectric relaxation spectroscopy (DRS) spectra of lyophilized horse myoglobin powder. Real part (permittivity) and imaginary part (dielectric loss) are presented. Reproduced with permission from [140], ACS, 2009.

**Figure 13 pharmaceutics-13-00534-f013:**
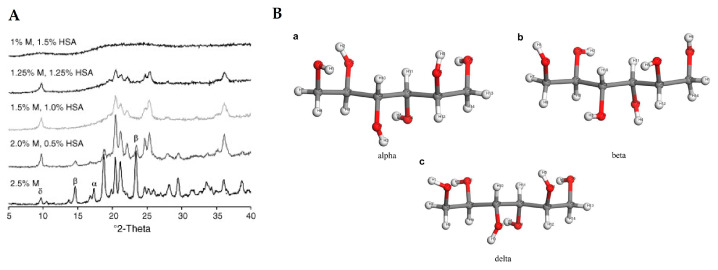
XRD analysis of lyophilized human serum albumin (HSA) formulations with different amounts of mannitol (M) is depicted in figure (**A**). Reproduced with permission from [153], Elsevier, 2006. In figure (**B**) the three forms of mannitol (a—alpha, b—beta and c—delta) are depicted in structural arrangements. Reproduced with permission from [154], Elsevier, 2020.

**Figure 14 pharmaceutics-13-00534-f014:**
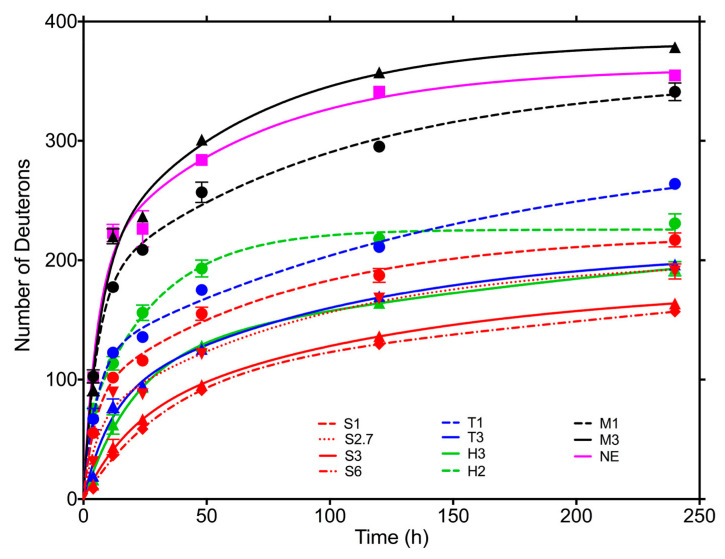
Time dependence of deuterium exchange for lyophilized monoclonal antibody in several different formulations (exposure to D_2_O vapor at 11% RH and 22 °C). Formulations: M3, mannitol 3:1; M1, mannitol 1:1; S6, sucrose 6:1; S3, sucrose 3:1; S2.7, sucrose 2.7:1; S1, sucrose 1:1; T3, trehalose 3:1; T1, trehalose 1:1; H3, histidine 3:1; H2, histidine 2:1; NE (no-excipient), without excipient. Ratios represent excipient/antibody w/w ratio. Reproduced with permission from [137], ACS, 2018.

**Figure 15 pharmaceutics-13-00534-f015:**
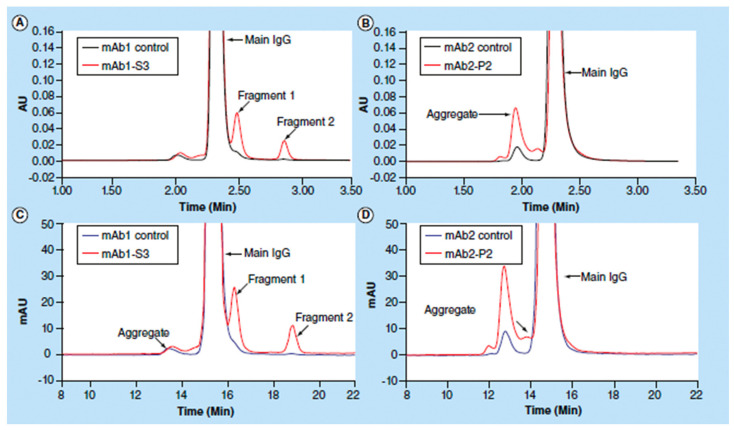
Size-exclusion chromatography (SEC) analyses of control monoclonal antibodies mAb1 and mAb2, antibody mAb1 after thermal stability studies for 3 months and an in-process antibody mAb2. Chromatograms (**A**,**B**) were obtained using UHPLC (ultra high-performance liquid chromatography), whereas chromatograms (**C**,**D**) were obtained by HPLC. Reproduced with permission from [185], Elsevier, 2018.

**Figure 16 pharmaceutics-13-00534-f016:**
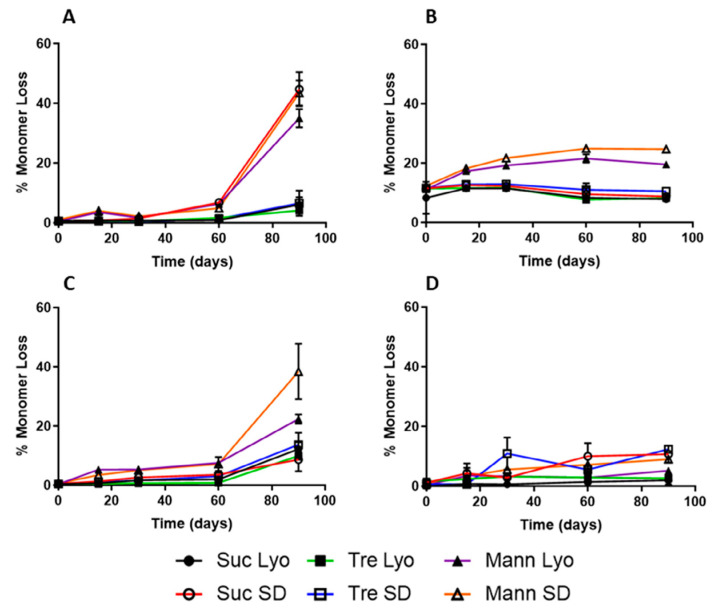
SEC studies on stability of samples containing myoglobin (**A**), BSA (bovine serum albumine) (**B**), β-lactoglobulin (**C**), or lysozyme (**D**). Formulations are lyophilized (Lyo) or spray-dried (SD) and have three sugar excipients (sucrose—Suc, trehalose—Tre, mannitol—Mann). Reproduced with permission from [175], Elsevier, 2019.

**Figure 17 pharmaceutics-13-00534-f017:**
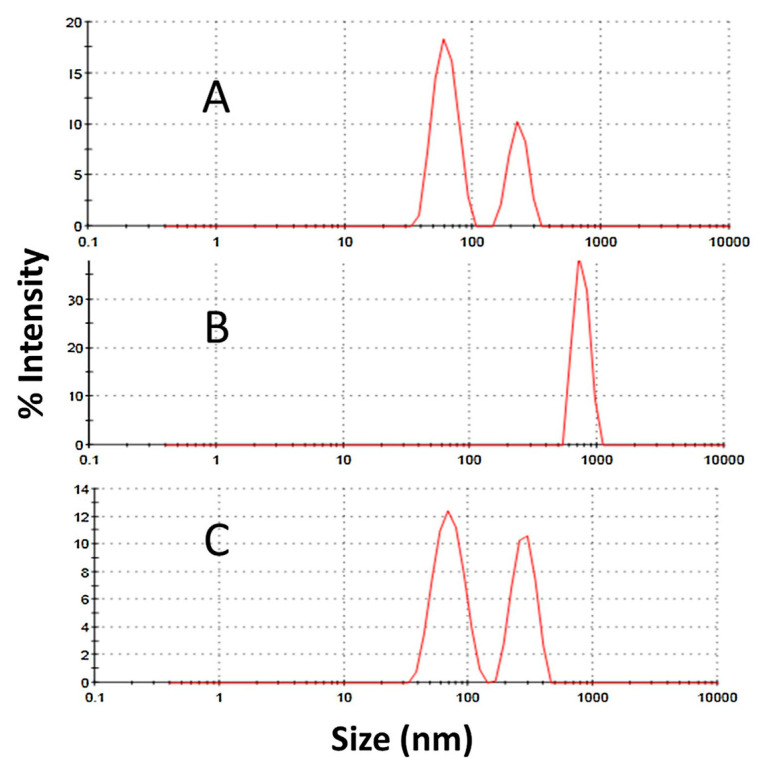
Dynamic light scattering (DLS) analyses for a desorbed quadrivalent human papillomavirus virus-like particles (HPV VLP) vaccine at different pH values: (**A**) at pH 6.2, (**B**) at pH 4.0, (**C**) at pH 8.0. Data are shown as intensity size distribution. Aggregate formation by forced degradation conditions can be observed in Figures (**B**,**C**), pH 4.0 and 8.0, respectively. Reproduced with permission from [187], Elsevier, 2017.

**Table 1 pharmaceutics-13-00534-t001:** Characteristic amide band frequencies and their associated secondary structures [64].

Frequency of the Band (cm^−1^)	Amide Region	Vibrations	Type of 2nd Structure
1680 1670–1680 1650–1655 1640	Amide I	H-bonded C=O stretch	β-Turn
			β-Sheet and β-barrel
			α-Helix
			Loose β-sheet
1300–1340	Amide III	N–H and C–H bend	α-Helix
1260			Disordered
1235–1250			β-Sheet
930–950	Backbone	N–C_α_–C stretch	α-Helix

**Table 2 pharmaceutics-13-00534-t002:** The effect of sucrose and histidine as excipients on T_g_ is presented in the table. The moisture content was measured by Karl Fischer titration [137].

Formulations	pH	Sucrose/mAb Ratio (*w*/*w*)	Histidine/mAb Ratio (*w*/*w*)	Moisture Content	T_g_
5H2	5		2:1	2.82 ± 1.31	94
5S1	5	1:1		2.72 ± 0.24	ND
5S1H0.5	5	1:1	0.5:1	1.67 ± 0.77	90
5S1H1	5	1:1	1:1	2.57 ± 0.01	90
6H2	6		2:1	2.79 ± 0.67	106
6S1	6	1:1		2.01 ± 0.22	93
6S1H0.5	6	1:1	0.5:1	1.60 ± 0.39	95
6S1H1	6	1:1	1:1	1.95 ± 0.38	101
H2	6.8		2:1	0.99 ± 0.34	105
S1	6.8	1:1		1.10 ± 0.03	ND
S1H0.5	6.8	1:1	0.5:1	2.55 ± 0.13	94
S1H1	6.8	1:1	1:1	2.12 ± 0.76	98

**Table 3 pharmaceutics-13-00534-t003:** Overview of the methods discussed above with some.

Method	Physical Principles	Sensitivity	Potential Damages	Pros	Cons
FTIR	C=O, N–H and C–N vibrations are measured as absorption or emission due to infrared light	Low to medium (on a global level)	Protein can be damage if too much pressure is applied when analyzing in ATR mode	Fast measurement, easy setup, non-expensive equipment, no hazardous chemicals are used, with ATR mode the sample can be recovered, small amount of sample needed	Only secondary structure can be evaluated, if KBr pellet mode is used, the sample cannot be recover, only analysis on a global level can be done and cannot provide site-specific information on specific portions of the protein sequence and their interactions with excipients, often unable to detect subtle structural differences
NIR	C=O, N–H and C–N vibrations are measured in the near-infrared region	Low to medium (on a global level)	No damages	Fast analysis, small amount of sample needed, no inert gas purging, easy setup and non-expensive equipment	Only secondary structure on a global level can be evaluated, water can interfere within protein spectra in some cases, cannot provide site-specific information on specific portions of the protein sequence and their interactions with excipients, often unable to detect subtle structural differences
Raman	C=O, N–H and C–N vibrations are measured as inelastic scattering after light excitation	Medium to high (on a global level)	Samples are usually damaged due to laser light irradiation	Very small amount of sample needed,	Samples cannot be recovered, longer time needed for measurements, more difficult equipment setup, only global level analysis
UV–Vis	Displacement of absorption (of UV or visible light) peaks is measured	Low to medium (on a global level)	No damages	Easy and non-expensive equipment setup, samples can be recovered, fast analysis	Only global level analysis of tertiary structure, cannot provide site-specific information on specific portions of the protein sequence and their interactions with excipients, often unable to detect subtle structural differences
Fluorescence	Emission of residual aromatic amino acids is measured after absorption of light or electromagnetic radiation	Medium (on a local level)	No damages	Tertiary structure on a local level, intensity and peak maxima position can be measured	Higher amount of sample needed, sample preparation for measurement is crucial
CD	Difference in absorbance is measured, involving circularly polarized light (left- and right-handed light)	Medium (on a global level)	No damage (except when temperature dependence experiment is applied)	Secondary and tertiary structure can be analyzed, small amount of sample needed	Only global level analysis, if temperature dependent experiment is applied the sample cannot be recovered, nitrogen gas purging needed
ssNMR	^1^H, ^13^C, ^15^N chemical shifts are measured after magnetic field excitation of the nuclei sample	High (on a global and local level)	No damage	Conformation and dynamics can be measured on a global and local level, different nuclei can be analyzed (proton, carbon, nitrogen)	Expensive equipment, long measurement time, higher amounts of sample are needed, only in some cases the samples can be recovered
DSC	Change in heat capacity at T_g_ is measured (the difference in the amount of heat required to increase the temperature of a sample and reference is measured as a function of temperature)	High to medium (on a global level)	Sample is damaged and cannot be recovered	Conformational changes and crystallinity of the sample can be evaluated	Sample cannot be recovered, only global level analysis, necessary to have well-characterized drug compounds
DRS	Rational motions of dipole-bearing groups are measured (as a function of frequency)	High to medium (on a global and local level)	Sample can be damaged	Global and local analysis	Sample usually cannot be recovered, higher amounts of sample are needed, difficult sample preparation, water may interfere with the sample analysis
XRD	Crystal structure is evaluated by irradiating the sample material with X-rays and measuring the intensities and scattering angles that leave the material	High (on a global level)	Samples can be damaged due to X-ray irradiation	Conformation and crystallinity of samples can be evaluated	Only global level analysis, sample usually cannot be recovered, expensive equipment
ssHDX-MS	Amide hydrogen exchange with deuterium in solid (by exposure to D_2_O) is measured with LC–MS	High (on a global and local level)	Samples are damaged due to deuteration and MS analysis	Very good correlation with aggregates formation and physical stability on storage, when peptide digestion is employed the samples can be analyzed on a local level providing also site-specific information on interactions between the protein and excipients	Samples cannot be recovered, difficult setup and expensive equipment
SEC	Chromatographic method in which molecules are separated by size, and in some cases molecular weight	High (on a global level)	Can be damage in some cases by mobile phase or column	Standard method for aggregation studies, small amount of sample is needed, fast method if automation is employed, also degradation products can be detected	Expensive equipment, some particles may not be detected or separated
DLS	Measures the Brownian motion of macromolecules in solution that arises due to bombardment from solvent molecules, and relates this motion to the size of particles to determine their size distribution	High (on a global level)	If temperature dependence or zeta-potential is measured, samples are damaged (aggregated)	Can analyze particles that may not be seen with SEC, easy setup and non-expensive equipment, fast analysis	Cannot differentiate molecules that are closely related (monomer and dimer) since it is a low- resolution method, it must be used on highly dilute solutions, restricted to transparent samples, very sensitive to temperature and solvent viscosity

Abbreviations: FTIR—fourier transform infrared; NIR—near-infrared; CD—circular dichroism, ssNMR—solid-state nuclear magnetic resonance; DSC—differential scannign calorimetry; DRS—dielectric relaxation spectroscopy; XRD—X-ray diffraction; ssHDX-MS—solid-state hydrogen-deuterium exchange mass spectrometry; SEC—size-exclusion; DLS—dynamic light scattering; ATR—attenuated total reflectance; T_g_—glass transition temperature; LC-MS—liquid chromatography with mass sepctrometry; MS—mass spectrometry.

**Table 4 pharmaceutics-13-00534-t004:** Comparison of different methods used for analysis of the same or similar protein sample.

Method	Pros	Cons	Information	Comparison with Other Methods
FTIR	Secondary structure determination, samples can be recovered, small amount of sample needed.	Structure determination only on a global level, poor correlation with other methods and especially SEC.	Amide I region was analyzed for each formulation and used to compare protein secondary structure. Little difference was observed with either changes in excipient or processing conditions, with the exception of β-lactoglobulin, where for spray-dried samples an increase in the heterogeneity can be deducted.	In comparison to ssHDX-MS, the results were relatively inconsistent and poor correlation was observed with results from SEC analysis. On the other hand is the only method in this study to characterize proteins secondary structures in solid. Further, is a fast and routinely analysis.
Fluorescence	Tertiary structure determination, samples can be recovered.	Poorer correlation with stability studies than with the ssHXD-MS. Measurements are not possible with lower concentrations.	Changes in tertiary structure correspond to shifts in the peak. Fluorescence spectra showed process related differences for BSA, they may be attributed to hydration differences, since spray-dried samples have lower moisture content. Lysozyme samples showed significant differences in peak position that is depended on formulation and processing conditions. Mannitol-containing formulations displayed red shifts, whereas sucrose samples displayed the blue ones. No difference was observed with trehalose samples.	In comparison to ssHDX-MS has weak correlation with long-term storage stability. On the other hand is quicker and has an easy equipment setup. Similar to FTIR is the only method for tertiary structure characterization, which is relatively fast and routinely.
XRD	Only method with DSC for sugars crystallization analysis.	Information only on global level with no sample recovery.	Formulations containing sucrose or trehalose were all completely amorphous, whereas mannitol samples showed minor peaks on XRD, indicating the presence of crystalline mannitol.	Mannitol samples showed some crystallization, which was observed also with DSC. The mannitol crystallization might be reflected in poorer storage stability, which was confirmed with ssHDX-MS and SEC.
DSC	Together with XRD analyzes and confirms samples crystallinity, as well as measures samples T_g_, which can be compared.	Samples cannot be recovered, the information only on a global level only.	T_g_ values were determined for sucrose and trehalose formulations, whereas T_m_ was determined for mannitol formulations. Sucrose samples had lower T_g_, whereas trehalose samples showed process-dependent differences in T_g_, with higher values for spray-dried samples. For samples with mannitol, the T_m_ confirmed crystallinity.	The results do not correlate good with storage stability measured by SEC, except for mannitol samples, which have shown to be crystalline and therefore less stable on long-term. Mannitol crystallization was observed also in XRD analysis.
ssHDX-MS	Good correlation with storage stability, analysis on global and local level (if peptide digestion is employed).	Expensive and rather complex equipment. Samples cannot be recovered after analysis. Longer times require for the experiments.	Mannitol formulations showed greater deuterium uptake and, hence, decreased storage stability (probably due to phase separation caused by the crystallization of the excipient), which correlates great with highest aggregate content measured by SEC. Similar results were obtained with either the deconvoluted peak area or the maximum deuterium incorporation.	In contrast to FTIR and fluorescence, ssHXD-MS gives a very good and consistent correlation with aggregation studies with SEC. On the other hand, the method requires much longer times for the analyses and it is not yet a routinely measurement.
SEC	Very good and reliable (standard) method for aggregation analysis and storage stability studies.	Measurements are done in solution—conditions only after the reconstitution of proteins; more expensive equipment.	The percentage of aggregates was greatest in mannitol formulations containing mannitol, with the exception of myoglobin spray-dried with sucrose and all formulations of lysozyme (spray-dried samples had greater aggregate content).	Provides information on long-term stability as measures the loss of monomer (aggregates formation). Despite ssHDX-MS, that can predict aggregation to certain degree due to good correlation with storage stability, SEC is the only method to really measure the extent of aggregation that occurred within the samples.

Abbreviations: FTIR—fourier transform infrared; XRD—X-ray diffraction; DSC—differential scannign calorimetry; ssHDX-MS—solid-state hydrogen-deuterium exchange mass spectrometry; SEC—size-exclusion.

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
