# Peer review of "Analytical Techniques for Structural Characterization of Proteins in Solid Pharmaceutical Forms: An Overview"

_pharmaceutics, 2021, doi:10.3390/pharmaceutics13040534_

Round 1
Reviewer 1 Report
In the manuscript the authors give a review of analytical techniques for structural characterization of proteins in solid pharmaceutical forms: in my opinion the review is well designed and exhausitve.
Author Response
We thank the Reviewer for the positive opinion.
Reviewer 2 Report
The review article by Bolje and Gobec is a comprehensive presentation of an array of different biophysical methods used to evaluate the stability of proteins in solid state and potential aggregation under different formulations. The description of typical results using different methods is presented and the comparison among them is briefly mentioned. I would suggest the following changes before publication.
1) Comparison of different methods in sensitivity and potential damages to samples should be tabulated.
2) Comparison of different methods in analyzing proteins in the same or very similar formulations should be tabulated and cross-validated.
3) Pros and Cons for different methods should be tabulated.
4) The physical principles for different methods can be tabulated and compared.
5) The description of protection of different formulations is too superficial. Please add more physical or molecular details on the protection principles.
6) Other minor questions are listed here.
Ln 146-148, what is the region amide III here?
Fig. 2, region II (1550 cm-1) was not presented, but discussed in the paragraph under it.
Ln 223-225, what is typical optical path length used here?
Ln 214-215, NIR is said to be advantageous over Raman, but not discussed later in the section.
Ln 216-224, FTIR seems to reveal more information than NIR, although experimentally it requires more operations and steps in data analysis. Please clarify why NIR is better.
Fig. 6, the change in the 1100 cm-1 peak is significant. How much materials used? How sensitive is it?
Fig 10, the red color appears discolored. The changes in far UV plots are very small; can it be expanded to show them better?
Fig 11, the changes in T1p is very random and not significant. Why is it claimed to be good to measure differences? What is the difference between T1p and T1?
Ln 512-518, what is Tg? Is it related to Tm?
Ln 525-526, what is eutectic temperature?
Ln 549-550, is it 10^-5 to 10^11 Hz? DRS is unfamiliar to most of readers. It is better to put more detail on how it can provide information about structures and dynamics.
Ln 612, the three forms of manitol may be depicted in structural arrangements in a Figure. Which form is better in protection?
Fig. 15, the blue background should be removed.
Page 25, the conclusion is very vague in comparison among different methods, especially about pros and cons and selection of different methods.
Author Response
1) Comparison of different methods in sensitivity and potential damages to samples should be tabulated.
Our reply: This comparison was done and is presented in a new table, Table 3.
2) Comparison of different methods in analyzing proteins in the same or very similar formulations should be tabulated and cross-validated.
Our reply: A comparison of different methods in analyzing proteins in the same or very similar formulations (extracted from the same study, i.e. reference) was tabulated and is presented in a new table, Table 4.
3) Pros and Cons for different methods should be tabulated.
Our reply: Pros and cons for different methods were tabulated and are presented in a new table, Table 3.
4) The physical principles for different methods can be tabulated and compared.
Our reply: The physical principles for different methods were tabulated and are presented in a new table, Table 3.
5) The description of protection of different formulations is too superficial. Please add more physical or molecular details on the protection principles.
Our reply: The protection description was revised and improved. More details on the protection principles were added in the text on page 2. Some new refrences for this chapter were added.
6) Other minor questions are listed here.
Ln 146-148, what is the region amide III here?
Our reply: The amide III region is around 1300 cm-1. We added the region as number in the text.
Fig. 2, region II (1550 cm-1) was not presented, but discussed in the paragraph under it.
Our reply: We modified the position of Figure 2 citation in the text, as deals ony with amide I region. We also added the reference for Figure 2 after the citation fo this figure.
Ln 223-225, what is typical optical path length used here?
Our reply: The typical optical path length used for protein characterization is around 1 mm. However, path lenghts from 0.5 mm up to 10 mm are used. This information was added in the text.
Ln 214-215, NIR is said to be advantageous over Raman, but not discussed later in the section.
Our reply: NIR measurements are much more quicker. There is also no need for inert gas purgment and also the sample is preserved with no damage at all (Raman, since uses laser light, usually damages proteins in samples). The advantages of NIR specstroscopy over Raman were added in the text.
Ln 216-224, FTIR seems to reveal more information than NIR, although experimentally it requires more operations and steps in data analysis. Please clarify why NIR is better.
Our reply: NIR spectroscopy is better over FTIR since does not need inert gas purgment and therefore the equipement is easier to install. Further, NIR can be used to determine information on crystallinity of sampels, as well as the residual water content (no need for other methods to be used to determine water content, such as Karl Fischer, that also damages protein samples). NIR measurements are also quicker than FTIR. All this is already included in the text, therefore we did not add anything to the text to avoid duplication of information.
Fig. 6, the change in the 1100 cm-1 peak is significant. How much materials used? How sensitive is it?
Our reply: The concentration of the sample accordingly to the reference was 9 mg/mL and aprox. 6.5 mg of sample (material) was used in each experiment. The material is relatively highly sensitive in this region, since this region is directly related to some amino acids side chains and correlates good with protein unfolding, for example upon heat-denaturation. When a protein with a unprotected non-native structure (without excipients such as sucrose for example) is analyzed the changes can be significant in comparison to a protein with a native or potected structure. Additional informations were added in the figure caption, as well as in the text about this figure.
Fig 10, the red color appears discolored. The changes in far UV plots are very small; can it be expanded to show them better?
Our reply: We cannot modify the picture, since we have Copyright from Elsevier for the exact figure, that was downloaded from the Journal website as high-resolutiom image. Therefore we cannot either change the red color aor expand the far UV changes to show the better.
Fig 11, the changes in T1p is very random and not significant. Why is it claimed to be good to measure differences? What is the difference between T1p and T1?
Our reply: Despite the changes in T1p can be random and not significant as seen in the Figure 11 from the literature, in many cases can deliver information (predict) on long-term stability of protein formulation in solid state (correlation between relaxation times and stability – lower relaxation time values means greater stability). Thus, the relaxation times are directly related to molecular motion frequencies. Both T1p and T1 are spin – lattice relaxation times. T1p is measured in a rotating frame, whereas the T1 is measured in a laboratory frame. These insights were added in the text.
Ln 512-518, what is Tg? Is it related to Tm?
Our reply: The symbol Tg means glass transition temperature. The symbol is explained in the introdution at page 3. However, we added the explanation of the symbol also in the section on DSC to make it more clear and easier to understand to the reader.
Ln 525-526, what is eutectic temperature?
Our reply: The eutectic temprature is the lowest possible melting temperature over all of the mixing ratios for the involved component species. We added this explanation in the text.
Ln 549-550, is it 10^-5 to 10^11 Hz? DRS is unfamiliar to most of readers. It is better to put more detail on how it can provide information about structures and dynamics.
Our reply: This typographical error was corrected in the text. We added some additional insights into the text on how DRS can provide informations about protein structures and dynamics.
Ln 612, the three forms of manitol may be depicted in structural arrangements in a Figure. Which form is better in protection?
Our reply: The three forms of mannitol were added in Figure 13B, as well as the Figure literature reference (Figure and permission were added to the corresponding zip files). D-Mannitol is used as excipient in protein formulations for stabilzation and native structure preservation. Usually, a mixture and not only one form is used. Very important is enough mannitol that has to be used for proper stability, as well as to avoid any possible crystallization of the sugar.
Fig. 15., the blue background should be removed.
Our reply: We cannot modify the picture, since we have Copyright from Elsevier for the exact figure, that was downloaded from the Journal website as high-resolutiom image.
Page 25, the conclusion is very vague in comparison among different methods, especially about pros and cons and selection of different methods.
Our reply: The conclusion was revised. We added some comparison, as well as pros and cons and the selection of different methods.
Reviewer 3 Report
The paper entitled “Analytical Techniques for Structural Characterization of Proteins in Solid Pharmaceutical Forms: an Overview” is an interesting review of the analytical techniques suitable for characterizing protein powders, providing useful information in formulation development. The paper is well organized but it can improve further if the following points are properly addressed:
- Text on lines 76 to 100 in the introduction section and the text on lines 821 to 844 in the conclusions section present the same ideas. Please consider revising.
- The abbreviations for the techniques mentioned in lines 821 to 844 in the conclusions section were defined previously in the text.
- Please improve the quality/resolution of Fig. 15
- Table 2: It is not possible to understand the composition of the formulations studied in reference 128.
- Lines 33 to 35 in the introduction section: The need for solid pharmaceutical formulations is related to the limited stability of proteins in the solution. Thus this type of formulation is not a way to increase the stability of proteins in solution but an attempt to overcome this limitation. Please revise the sentence.
Author Response
- Text on lines 76 to 100 in the introduction section and the text on lines 821 to 844 in the conclusions section present the same ideas. Please consider revising.
Our reply: We changed the conclusions part of the manuscript, to suit the Reviewer’s comment.
- The abbreviations for the techniques mentioned in lines 821 to 844 in the conclusions section were defined previously in the text.
Our reply: The abbreaviations definitions were removed acordingly to the Reviewer`s comment.
- Please improve the quality/resolution of Fig. 15
Our reply: Figure 15 was changed with high-resolution image, downloaded from the reference (journal website). We have Copyright from Elsevier for the exact downloaded image. The image is already included in the zip file “Figures”.
- Table 2: It is not possible to understand the composition of the formulations studied in reference 128.
Our reply: Table 2 was modified to be more clear in terms of formulations compositions for sampels, that were studied.
- Lines 33 to 35 in the introduction section: The need for solid pharmaceutical formulations is related to the limited stability of proteins in the solution. Thus this type of formulation is not a way to increase the stability of proteins in solution but an attempt to overcome this limitation. Please revise the sentence.
Our reply: The sentence was revised acordingly to reviewr’s comment.
Round 2
Reviewer 2 Report
The points I raised in the last round have been addressed properly.